# Litters of Various-Sized Mummies (LVSM) and Stillborns after Porcine Reproductive and Respiratory Syndrome Virus Type 1 Infection—A Case Report

**DOI:** 10.3390/vetsci10080494

**Published:** 2023-08-01

**Authors:** Christine Unterweger, Heinrich Kreutzmann, Moritz Buenger, Eva Klingler, Angelika Auer, Till Rümenapf, Uwe Truyen, Andrea Ladinig

**Affiliations:** 1University Clinic for Swine, Department for Farm Animals and Veterinary Public Health, University of Veterinary Medicine, 1210 Vienna, Austria; heinrich.kreutzmann@vetmeduni.ac.at (H.K.); moritz.buenger@vetmeduni.ac.at (M.B.); andrea.ladinig@vetmeduni.ac.at (A.L.); 2Vetpraxis Hegerberg, 3053 Laaben, Austria; 3Institute of Virology, Department of Pathobiology, University of Veterinary Medicine, 1210 Vienna, Austria; angelika.auer@vetmeduni.ac.at (A.A.); till.ruemenapf@vetmeduni.ac.at (T.R.); 4Institute for Animal Hygiene and Veterinary Public Health, Faculty of Veterinary Medicine, University of Leipzig, 04103 Leipzig, Germany; truyen@vetmed.uni-leipzig.de

**Keywords:** porcine reproductive and respiratory syndrome virus, PRRSV-1, mid-gestation, SMEDI, LVSM, mummifications, stillborn piglets, prolonged gestation, PRRSV AUT15-33, modified live virus vaccine

## Abstract

**Simple Summary:**

In an Austrian piglet-producing farm, sudden occurrences of mummified foetuses of various sizes and stillborn piglets were observed, and date of births were delayed in more than 50% of the sows in the respective farrowing group. There are a few pathogens known to be involved in pathogenesis of this particular clinical picture, called SMEDI (short for “stillbirth, mummification, embryonic death and infertility”). All of them were included in the diagnostic work-up of three litters consisting of mummies and stillborns, but were not detected. Instead, high viral loads of the porcine reproductive and respiratory syndrome virus (PRRSV), a virus known to be involved in the aetiology of a variety of clinical signs in pigs, but not in the one of papyraceous mummifications, were found. This once more shows the variability of the clinical outcome of this pathogen.

**Abstract:**

Diverse origins and causes are described for papyraceous mummifications of porcine foetuses, but the porcine reproductive and respiratory syndrome virus (PRRSV) is not one of them. In contrast, PRRSV is unlikely to cause mid-term placental transmission but may cause late-term abortions and weakness of piglets. This case report describes a sudden occurrence of mummified foetuses of various sizes and stillborns and delayed birth (>115 days) in more than 50% of sows from one farrowing batch, while newborn piglets were mostly vital. Neither increased embryonic death nor infertility was reported. Three litters with mummies, autolysed piglets and stillborn piglets were investigated, and infections with porcine parvoviruses, porcine teschoviruses, porcine circoviruses, encephalomyocarditis virus, *Leptospira* spp. and *Chlamydia* spp. were excluded. Instead, high viral loads of PRRSV were detected in the thymus pools of piglets at all developmental stages, even in piglets with a crown–rump length between 80 and 150 mm, suggesting a potential mid-term in utero transmission of the virus. Genomic regions encoding structural proteins (ORF2–7) of the virus were sequenced and identified the virulent PRRSV-1 strain AUT15-33 as the closest relative. This case report confirms the diversity of PRRSV and its potential involvement in foetal death in mid-gestation.

## 1. Introduction

The acronym SMEDI describes the simultaneous occurrence of stillbirths, mummifications, embryonic deaths and infertility in gilts and sows and was initially introduced by Dunne et al. (1965) after isolating porcine teschovirus in association with these clinical signs [1]. In the meantime, the potential for triggering SMEDI has been demonstrated in a number of pathogens, of which porcine parvovirus 1 (PPV1) is the most common [2]. Further potential pathogens involved in the SMEDI aetiology primarily include viruses such as porcine circovirus type 2 (PCV2) [3], porcine sapelovirus [1] and encephalomyocarditis virus (EMCV) [4], but also bacteria such as Leptospira [5]. Mummifications are a result of foetal death after ossification. In pigs, the most common form is papyraceous mummification; foetuses are brownish or black in colour, dehydrated with sunken eyes and have a shrivelled appearance [6]. To attain this condition, foetal death has to take place before the onset of the immunocompetent phase (around gestation day 70). The gestation stage in which the foetus died can be estimated by measuring the crown–rump length (CRL), which should range between 40 and 150 mm during this period [7]. Later death results in either autolysed or macerated foetuses as well as stillborn piglets. Besides infectious causes, foetal mummification has also been associated with parity, litter size, uterine capacity, nutritional factors and environmental temperature [8,9]. 

The porcine reproductive and respiratory syndrome virus (PRRSV) is one of the most devastating pathogens in pig production worldwide, causing episodes of reproductive failure in gilts and sows as well as respiratory disorders in growing pigs [10]. The manifestation of PRRSV-related clinical signs in gilts and sows depends on the immunological status of the herd, the stage of gestation, the virulence of the PRRSV strain, the occurrence of co-infections and other environmental and management factors [11]. An infection in the early phase of gestation can lead to embryonic death and, as a consequence, to an increased return-to-oestrus rate [12,13]. Infection of gilts and sows in the last trimester of gestation causes transplacental infection of foetuses, which has been demonstrated for both PRRSV-1 and PRRSV-2 in experimental studies [14,15,16]. As previously reported for PPV1 and PCV2, inter-foetal transmission of PRRSV is likely to occur since the status of adjacent foetuses is closely associated with thymic viral load and the likelihood of foetal death [17]. Consequently, the occurrence of late-term abortions, dead and weak-born piglets can be seen [18,19]. However, according to the literature, the virus is unlikely to cross the maternal–foetal interface and to cause clinical symptoms around the 45th day of gestation (mid-gestation), although the foetuses are generally susceptible upon direct in utero infection [19,20]. 

This case report describes the occurrence of mummified foetuses and autolysed and stillborn piglets alongside vital piglets and prolonged gestation in a conventional Austrian farm with a history of PRRS. High PRRSV viral loads were detected in the mummified foetuses, which were associated with a potential PRRSV infection in mid-gestation. 

## 2. Case Presentation

### 2.1. Anamnesis and Physical Findings

The case herd was a commercial, family-owned piglet-producing farm with 160 sows (Landrace × Large White) producing at five-week batch-farrowing intervals, located in Lower Austria. Piglets were sold directly after weaning at an age of four weeks except for replacement gilts. After an acute PRRS outbreak three years earlier, vaccination of the whole herd with a modified live virus (MLV) vaccine (sows: ReproCyc^®^ PRRS EU, Boehringer Ingelheim Vetmedica GmbH, Ingelheim am Rhein, Germany; piglets: PRRSFlex^®^, Boehringer Ingelheim Vetmedica GmbH, Ingelheim, Germany) was carried out. After subsequent continuous vaccinations of sows every four months and piglets prior to weaning, the farm was classified as PRRSV-positive stable with vaccination according to the definitions of Holtkamp et al. [21]. Additionally, sows were vaccinated against PPV1 and *Erysipelothrix rhusiopathiae* (Parvoruvac^®^, Ceva Santé Animale, Libourne, France) during the suckling period as well as against Influenza A virus (Respiporc FLU3^®^, Ceva Santé Animale, Libourne, France) around gestation day 70 according to the manufacturers’ recommendations. To prevent neonatal piglet colibacillosis and clostridiosis, vaccination (Enteroporc COLI AC^®^, Ceva Santé Animale, Libourne, France) was applied to sows three weeks prior to the calculated farrowing date. In addition to the vaccination against PRRSV, piglets were routinely vaccinated against *M. hyopneumoniae* and PCV2 (Porcilis^®^ PCV M Hyo, Intervet International B.V., Boxmeer, The Netherlands) before weaning. Prior to the first insemination, gilts were also vaccinated against PRRSV, Influenza A virus, PPV1 and Erysipelas starting at around 180 days of life, with the same vaccines as used in sows. Semen was purchased mainly, but not exclusively, from two different official boar stations, both negative for PRRSV. Determination of pregnancy was performed via transcutaneous ultrasonographic examination around gestation day 28. Five days prior to farrowing, sows were moved to the cleaned and disinfected farrowing units where they were kept individually. External and internal biosecurity included strict hygiene and disinfection controls over vehicles, equipment and personnel entering the farm as well as all-in/all-out and proper cleaning and disinfection within the barn. 

In October 2021, an obvious increase in litters with differently sized mummies and autolysed foetuses, as well as stillborn piglets, was noted by the farmer (group 1, farrowing in calendar week (cw) 42/21, Figure 1). 

The herd veterinarian suspected an outbreak of parvovirosis and therefore changed the PPV1 vaccine (Porcilis^®^ Ery + Parvo, Intervet GesmbH, Vienna, Austria) as a first consequence, but no diagnostic investigation was initiated. Six weeks later, in the next farrowing batch (group 2, farrowing in cw 48/21) with 29 sows and 7 gilts, 17 farrowings were delayed (>115 days) and 17 sows farrowed on gestation day 114 or 115. One sow had already aborted around gestation day 80, and another sow with confirmed pregnancy according to ultrasonography was not pregnant anymore at the predicted time of birth. These two sows were successfully inseminated at the next possible date. The average number of mummies/autolysed foetuses (11.9%; eight litters with ≥4 mummies/autolysed foetuses) in group 2 even exceeded those of group 1 (Figure 1). As neither increased infertility nor embryonic death was part of the problem, the commonly used term SMEDI does not describe this issue properly. Consequently, we introduce the term “litters of various sized mummies” (LVSMs). Live-born piglets were mostly vital, and 12.8 piglets per litter (93.6% of live-born piglets) were weaned in this farrowing group (Figure 2).

### 2.2. Diagnostic Methods and Laboratory Findings

Post-mortem examination of the mummified and autolysed foetuses as well as stillborn piglets of three litters (A, B, C; Table 1) from farrowing group 2 was conducted at the University Clinic for Swine, Vetmeduni, Austria (Table 2). The CRL varied from 100 mm to 280 mm in litter A, from 60 mm to 265 mm in litter B (Figure 3) and from 80 mm to 300 mm in litter C. 

Specific foetal organs were pooled as shown in Table 2; all foetal organ material within litter A and within litter B was pooled. Litter C foetuses were subdivided into two pools, one consisting of mummified foetuses with a CRL of 80–150 mm (C1, *n* = 6) indicating an age between 48 and 71 days of gestation and the other one consisting of autolytic foetuses/stillborn piglets with a CRL ≥ 200 mm (C2, *n* = 8) indicating a minimum age of about 90 days.

For the detection of PPV1, pooled samples of the placenta, liver, lung, kidney and umbilical cord were collected from each litter, and qPCR [22] was performed at the Institute of Virology, Vetmeduni, with negative results (Table 3). These results were confirmed by two other PCR protocols, a conventional PCR and a real-time PCR protocol [23,24], at the Institute for Animal Hygiene and Veterinary Public Health, University of Leipzig. At the same time, three cell lines (STE, SPEV and pk15) were inoculated with samples of the same organs to potentially isolate parvoviruses or other viruses, with negative results. PCV2 qPCR [25] as well as PCV3 qPCR [26] and encephalomyocarditis virus (EMCV) RT-PCR [27] from the foetal hearts also gave negative results (Institute of Virology, Vetmeduni Vienna, Austria). Neither porcine teschovirus (PTV) nor porcine enterovirus (PEV) was detected in the same pool used for PPV1 detection from the three litters via RT-PCR [28]; neither *Leptospira* spp. [29] nor *Chlamydia* spp. [30] were present in pools of the lung, liver or kidney. To exclude the involvement of PRRSV, thymus pools were analysed via RT-qPCR [31] with positive results. Positive samples were quantified with an in-house RT-qPCR targeting ORF1 [32], which resulted in 6.6 × 10^7^ genome equivalents [GE]/g tissue in litter A and 2.3 × 10^7^ GE/g tissue in litter B. The thymus pool sample C2 (autolysed foetuses, stillborn piglets) contained 3.3 × 10^10^ PRRSV GE/g tissue, and the thymus pool sample C1 (mummies) contained 2.4 × 10^8^ PRRSV GE/g tissue.

### 2.3. Further Steps and Outcome of the Case

Since the sows were vaccinated against PRRSV with an MLV, sequencing of the structural-protein-encoding region (ORF2–7) was performed. The detected PRRSV strain showed 95.5% shared nucleotide identity with the PRRSV-1 field strain AUT15-33 (GenBank acc. no. MT000052.1) as the best match in BLAST^®^ [33]. The vaccine strain was not detected. 

Regarding the PRRS vaccination strategy, nothing was changed. In the next farrowing group five weeks later, one gilt and one fifth-parity sow showed the same clinical outcome in terms of litters of various sized mummies, but afterwards the prevalence of mummies stabilised at lower than 4% (Figure 1). All sows except those that would have been eliminated anyway due to a high parity number were successfully inseminated with a farrowing rate of more than 90%. Once more, 15 weeks later, there was a peak of 12.2% stillborn piglets, but no increase of mummifications (Figure 2). 

## 3. Discussion

This case report describes the occurrence of mummified, autolysed and stillborn foetuses and piglets, a clinical presentation usually linked to PPV, PTV, PCV2 and other pathogens, but not particularly to PRRSV. Still, PRRSV was the only pathogen found in the delivered foetal material, which was not expected, especially not in mummified piglets, resulting in a closer investigation of this case. 

What is unusual about the current case report in terms of PRRSV is the fact that (1) the CRL of numerous mummies suggested foetal death in mid-gestation; (2) piglets born alive were mostly vital, with no increased suckling piglet losses; (3) no early farrowing but prolonged gestation occurred; and (4) the litters showed distinct mummies in various sizes. 

PRRSV is transmitted via the maternal–foetal interface from viraemic sows to foetuses, but effective placental crossing of the virus is temporally restricted to the last trimester of gestation, resulting in late-term abortions or birth of dead and/or weak piglets [18,19,34]. Prolonged gestation lengths are usually not associated with PRRSV infections but are correlated with all SMEDI-inducing pathogens, which are transmitted very slowly from one foetus to another within the uterine horns—too slow to initiate abortion. Due to the low number of vital piglets, birth induction may be delayed. In the present case, this might also have been the reason for the overdue birth dates; however, it remains unclear why PRRSV infection of the foetuses occurred at an earlier stage of gestation, during which vertical transmission of the virus from the sows to the foetuses is not commonly described. 

The detected virus was most closely related to a frequently diagnosed and well-described field virus strain in Austria, namely PRRSV AUT15-33 [13]. As seen in a previous infection study with PRRSV AUT15-33 in the last trimester of gestation, the number of infected foetuses after 21 days was highly variable, from very few to all [16]. The speed of spread is therefore likely to vary significantly in the uterus of individual sows. In the present case, the virus must have spread very slowly within the uterus, which can be concluded based on the different sizes and developmental stages of foetuses at the time of death and the absence of abortions during mid-gestation. It is not clear which factors influence the speed of in utero spread, but it could be speculated that the PRRS status of the sows at the time point of infection is crucial. In the trial mentioned, naïve gilts were infected [16], whereas in the present case report, the sow herd had been previously immunised with an MLV vaccine. This difference might be important for the development of the different disease patterns. PRRSV AUT15-33 might behave differently in pregnant sows than the isolates used previously to infect sows in mid-gestation [19,20]. However, an experimental trial with pregnant gilts and PRRSV AUT15-33 infection in mid-gestation would be needed to support this statement. Since Christianson et al. (1993) used a PRRSV-2 isolate and Kranker et al. (1998) used a PRRSV-1 isolate for mid-gestation infection, the PRRSV species might not be the decisive factor. 

Generally, vaccination with an MLV vaccine is efficacious in sows and provides partial protection against challenge with heterologous PRRSV-1 isolates [35,36,37] and at the same time improves the reproductive performance compared to unvaccinated challenged animals [38]. A key point for the efficacy of MLV vaccines in the field depends on the vaccination strategy, including the time point and the interval between vaccinations, as well as biosecurity measures [39]. Both external and internal biosecurity protocols are strictly followed in the presented farm. The risk of pathogen introduction is therefore rather low, yet PRRSV was able to enter the herd for a second time within three years. While after the first PRRSV introduction, a PRRS vaccine virus derivate circulated in the herd, the virus found in the latter case was most closely related to PRRSV-1 AUT15-33 [13]. In the affected herd described in the case report from Sinn et al., a return-to-oestrus rate of 60% in sows was observed. Additionally, up to 50% of piglets were stillborn, and another 40% of piglets died within the first days of life in one farrowing group [13]. In comparison, piglets born alive in the presented case report were not weak-born, and losses during the suckling period were extremely low. It remains unclear why the clinical symptoms in the two farms differed so much; one explanation might be that in contrast to the presented case, the herd in the cited case report was PRRSV-naïve prior to the outbreak, indicating a beneficial effect of the MLV vaccine (Sinn et al. 2016). Another explanation might be that the virulence of the PRRSV strain was reduced by changes of 5–6% in ORF2–7 over time. 

Even though not all four clinical presentations included in the term SMEDI are always present or noticeable, the term is still commonly used. In the present case, neither increased embryonic death nor increased infertility was detectable; therefore, the term SMEDI does not accurately describe the clinical presentation. In order to define the clinical presentation of the different developmental stages of the foetuses at the time of death, we created the term LVSMs (litters of various sized mummies). Since it is quite possible that the term “mummy” is used differently, it underlines the need for a precise description of the term and the measurement of objective parameters such as the crown–rump length. 

The finding of PRRSV in mummies does not necessarily prove that PRRSV is responsible for the clinical signs. We are aware that PRRSV might not be the only pathogen or factor involved in the pathogenesis of the present clinical outcome, even though no plausible pathogen was detected. Indeed, we focused extensively on accurate testing of all relevant “SMEDI” pathogens known in our region and, in addition, unusual causes like PTV/ PEV and EMCV were investigated. In particular, direct detection of PPV1 as well as other PPVs in diverse organ tissues was intensified by performing a culture experiment in addition to the routine PCR, but a negative result was found. Investigations of the reference organ tissues of three complete litters and corresponding placental tissues with negative results derived from validated reference methods gives credibility to the negative results, though. 

## 4. Conclusions

This case report shows once more the diversity of PRRSV. It also makes us aware that we might have to expect PRRSV to be involved in foetal deaths during various foetal developmental stages, including those in mid-gestation. LVSMs (litters of various sized mummies) might be a more precise term to describe the clinical signs than SMEDI, especially if no data about embryonic death and infertility are available.

## Figures and Tables

**Figure 1 vetsci-10-00494-f001:**
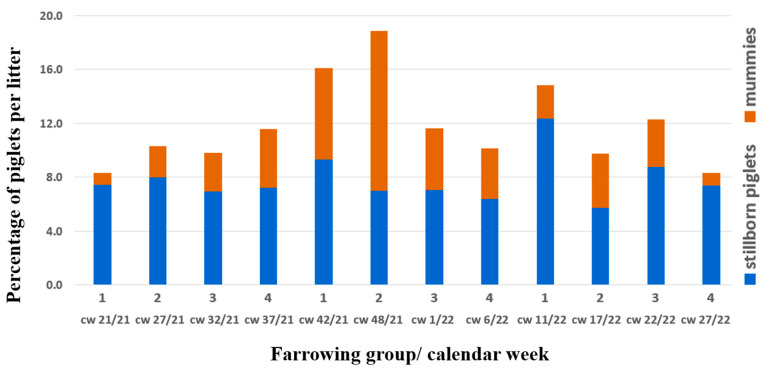
Stillborn piglets and mummies per litter in % over time; blue bars show the mean percentage of stillborn piglets, orange bars show the mean percentage of mummies per litter in the respective farrowing group and calendar week (cw). Foetal material for diagnostic work-up belonged to group 2 (cw 48/21).

**Figure 2 vetsci-10-00494-f002:**
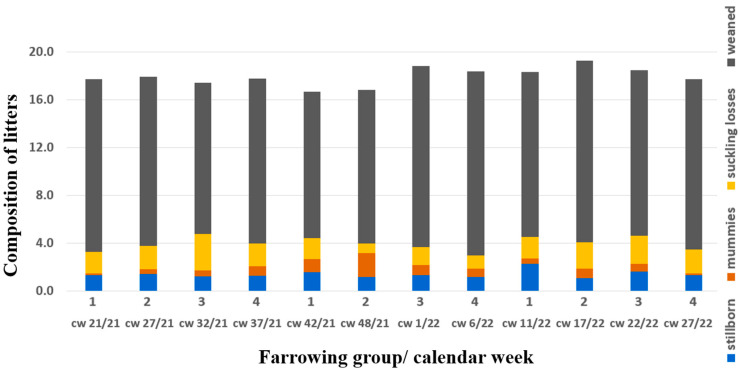
Composition of litters in absolute numbers over time; grey bars show the mean number of weaned piglets, yellow bars show the mean number of suckling losses (piglets that were live-born but died before weaning), blue bars show the mean percentage of stillborn piglets, orange bars show the mean percentage of mummies per farrowing group. Foetal material for diagnostic work-up belonged to group 2 (cw 48/21).

**Figure 3 vetsci-10-00494-f003:**
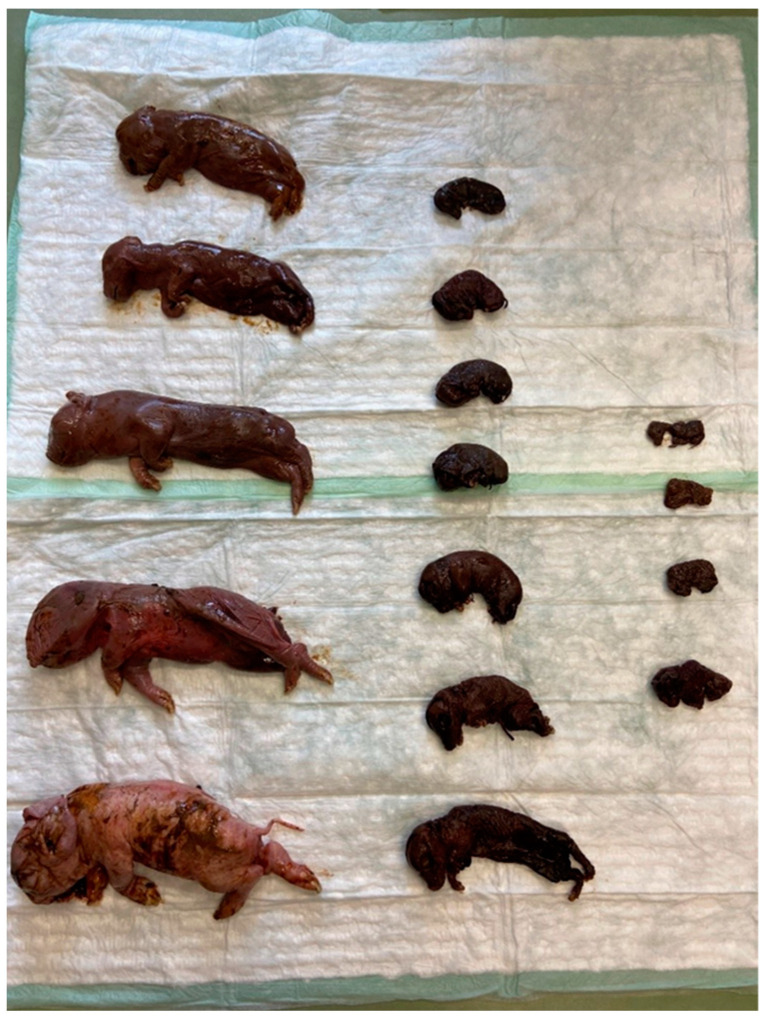
Different developmental stages within one litter (group 2, litter B). Six more piglets of this litter were live-born, vital and all weaned and healthy.

**Table 1 vetsci-10-00494-t001:** Reproductive data on sows of affected litters.

Sow ID	Parity	Nr of Total PRRSV Vaccinations *	Days of Gestation	Composition of Following Litters **		
				Live-Born	Mummies	Stillborns
Litter A	3	6	115	20	2	1
Litter B	2	5	115	18	0	2
Litter C	5	8	116	17	1	2

* Including basic immunisation; ** group 2 cw 17/22, as shown in Figure 1 and Figure 2.

**Table 2 vetsci-10-00494-t002:** Overview of submitted abortion material.

Litter	Crown–Rump Length in mm																Nr of Live-Borns *
A	**280**	**255**	**250**	265	230	210	130	100									8
B	**265**	**245**	**240**	200	185	175	135	135	130	125	110	95	90	70	60	60	6
C	300	290	255	220	220	210	205	200	150	130	120	110	80	80			5

Each line represents one litter (A–C). Each blue-coloured field symbolises one stillborn, but normally developed piglet; each yellow field symbolises one autolysed piglet; and each pink field symbolises one mummified foetus. Additionally, in each field, the crown–rump length of each individual is shown. All samples from animals within bold black borders were pooled for further diagnostics. * Number of additional live-born piglets in the respective litter.

**Table 3 vetsci-10-00494-t003:** Molecular investigations of foetal organs.

Pathogen	Tested Organ	Method	Result	PCR Assay According to
PPV1 PPV in general	Placenta, liver, lung, kidney and umbilical cord	PCR	Negative	[22]
		PCR	Negative	[23]
		qPCR	Negative	[24]
		Virus isolation	Negative	
PTV, PEV	Placenta, liver, lung, kidney and umbilical cord	RT-PCR	Negative	[28]
PCV2	Heart	qPCR	Negative	[25]
PCV3	Heart	qPCR	Negative	[26]
EMCV	Heart	RT-PCR	Negative	[27]
*Leptospira interrogans*	Lung, liver, kidney, placenta	PCR	Negative	[29]
*Chlamydia* spp.	Lung, liver, kidney, placenta	qPCR	Negative	[30]
PRRSV	Thymus	RT-PCR	**Positive**	[31]

## Data Availability

The data presented in this case report are available on request from the corresponding author.

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
