# Peer review of "Litters of Various-Sized Mummies (LVSM) and Stillborns after Porcine Reproductive and Respiratory Syndrome Virus Type 1 Infection—A Case Report"

_vetsci, 2023, doi:10.3390/vetsci10080494_

Round 1

Reviewer 1 Report

The manuscript reported a case of porcine reproductive and respiratory syndrome virus (PRRSV) infection resulted in litters of various sized mummies (LVSM) and stillborns in an Austrian piglet-producing farm immunized with a modified live virus vaccine. The authors suggested that PRRSV would be involved in foetal deaths during various foetal developmental stages including in mid-gestation. It provides useful information for prevention and control of PRRSV infection in pig production.

Author Response

The authors thank reviewer 1 very much for the positive feedback!

Reviewer 2 Report

In this article, the author describes a PRRSV-induced event characterized by the production of SMEDI. By excluding common pathogens that cause SMEDI, such as PPV1 and PCV2, it can be confirmed that this SMEDI event was caused by PRRSV infection, enriching the understanding of the clinical manifestations of PRRS. However, there are some issues that need further clarification.

1. The current data cannot actually prove the true time of virus infection. Based on previous experience, PRRSV cannot trigger SMEDI (or report) after infection, and the author has also pointed out that virus dissemination, virulence, and immune status of infected sows are related. Therefore, the authors should use a more detailed analysis to demonstrate "a potential mid-term in utero transmission of the virus."

2. The author declares that the pig farm has strict on-site and off-site biosecurity measures and also explains that PRRSV can still enter the site. Therefore, were the PRRS outbreaks in A, B, and C litters caused by the same time of infection, or were they progressive infections? Different forms of infection may yield different analyzes of the results under the same virus virulence and sow immune status.

3. More information is needed about the background of pigs in litters A, B, and C, such as the pregnancy times and the number of PRRS vaccinations after entering the pig farm, which may affect the clinical symptoms of PRRSV infection.

none

Reviewer 3 Report

This case report describes a sudden occurrence of mummified foetuses of various sizes and still borns, delay of birth (>115 days) in more than 50 % of sows from one farrowing batch, while newborn piglets were mostly vital.

  And this causative agent considered to be PRRSV infection, because other causative agents porcine parvoviruses, porcine tescho viruses, porcine, circoviruses, encephalomyocarditis virus, Leptospira spp. and Chlamydia spp. were excluded.

And this case report confirms the diversity of clinical signs caused by PRRSV and a potential involvement in foetal death in mid gestation.

But I cannot find this case report showed special sample of PRRS infection.

Therefore, please show clearly this case report is how important to publish in PRRSV infection.

Round 2

Reviewer 3 Report

You have immediate revised seriously and sincerely well. Therefore, it is the time to accept for publication in Veterinary Science.